# MACHINE LEARNING RESEARCH COMMUNICATION VIA ILLUSTRATED AND INTERACTIVE WEB ARTICLES

**J Alammar**
Arpeggio Research
ar.pegg.io
jay.alammar@pegg.io

## ABSTRACT

The recent explosion in machine learning research activity poses challenges for both researchers who aim to widely disseminate their work, as well as to readers who find it challenging to keep up with the onslaught of new research ideas. In this paper, we describe a workflow for creating a spectrum of machine learning research communication artifacts optimized to maximize the clarity of scientific communication, advance the fronts of explainability and interpretability, as well as empower the community to reproduce research software. The workflow describes creating a spectrum of communication artifacts including visuals, animations, interactive explorables, reproducible notebooks, open-source software, and software packages. The articles produced by this workflow have explained cutting-edge-ML research to a large audience and were read over three million times.

## 1 INTRODUCTION

Machine learning research has seen an explosion in the number of papers in recent years. Adapting to this boom time activity introduces challenges to both researchers who aim to make their work accessible to the broadest possible audience, as well as to readers who have to be judicious in allocating their attention to keep up with the onslaught of new research ideas and directions.

This paper shares a workflow summarizing practices, tools, and considerations taken in creating a series of web articles[1] which over the last few years has introduced a wide audience to advanced machine learning research concepts such as self-attention (Vaswani et al., 2017), the Transformer architecture, and the inner-workings of leading NLP models such as ELMo (Peters et al., 2018) and BERT (Devlin et al., 2019).

## 2 WORKFLOW TO CREATE A SPECTRUM OF COMMUNICATION ARTIFACTS

We present our workflow used for creating a spectrum of machine learning communication artifacts. In increasing sophistication and required time commitment, these artifacts are: the hero image (Section 2.1), the Twitter thread (Section 2.2), the illustrated/animated article (Section 2.3), the interactive article (Section 2.4), and interpretability research software (Section 2.5).

### 2.1 THE HERO IMAGE

Readers are exposed to a research paper in various contexts. Several of these contexts allow for, some are even improved by, an image representing the central concept of a paper. This is a key communicative opportunity that became even more important when driven by the covid-19 pandemic, machine learning conferences were conducted virtually and conference software started presenting the first image of a paper as a thumbnail for attendees browsing the accepted papers. Figure 1 shows notable and memorable hero images from Olah (2015) and Karpathy (2015).

---

[1]https://jalammar.github.io/

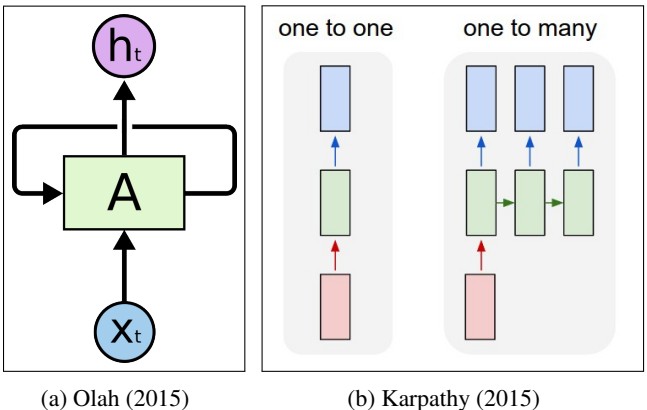

(a) Olah (2015)  (b) Karpathy (2015)

Figure 1: A hero image explains a central concept of a research work aiding understanding and recall of the paper and the concept.

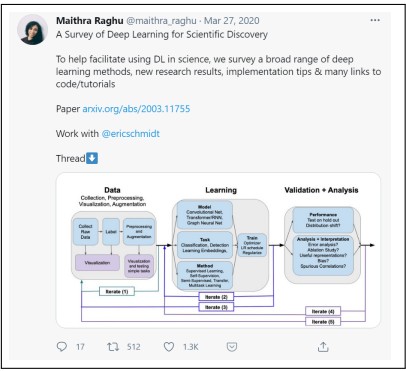

Figure 2: Crafting a brief summary of a paper's main ideas in three to eight tweets exposes a work to a wider audience.

Peyton Jones (2018) presented a heuristic indicating the title of a paper is read an order of magnitude more than the abstract (e.g., 1000 readers for the title vs. 100 readers for the abstract). The abstract itself is read an order of magnitude more than the other contents of the first page (e.g., 10 readers). The majority of correspondents to an informal and unscientific online poll [2] believed the hero image is seen by more people than those who read a paper's title.

In our workflow, a hero image emerges from the narrative created in an illustrated article (Section 2.3). Most complex machine learning ideas need more than one image to gently introduce a concept without overwhelming the reader with too much detail too early on. If, for example, a concept needs four images to set up and explain, the second or third image may serve as a reasonable hero image.

## 2.2 THE TWITTER THREAD

Twitter has increased in importance as a method to discover and share machine learning research. Crafting a summary of a paper's main ideas in three to eight tweets exposes a work to a wider audience. This is also a golden opportunity to use images and animations leading with the hero image. Figure 2 shows an example of the first tweet in a thread inviting readers to read Raghu & Schmidt (2020)[3].

In our workflow, we aim for the first tweet to motivate the concept and use an image to demonstrate the accessible visual style used in the remainder of the thread. It's common to link to the expanded research article or paper in the first and final tweets in the thread.

## 2.3 THE ILLUSTRATED/ANIMATED WEB ARTICLE

Next in the spectrum is a web article with a series of images and animations explaining a machine learning concept to a wide audience. Such an article can either be a work of research itself or exposition, explanation, or curation of existing research. Notable work in this category includes Karpathy (2015); Olah (2015); Britz (2015); Nielsen (2016); Ruder (2016); Rush (2018); Weng (2018); Vinyals et al. (2019) and LeCun & Misra (2021).

Beyond the basic form, what makes such work notable is how effectively it communicates advanced concepts to a wide audience that is not assumed to have read the research literature surrounding and leading up to the topic of the article. Such work is often created with empathy towards the reader, consideration of their background and purpose of reading, what is safe to assume they know, care to avoid jargon, and attention to the conceptual bite-size of each concept the article introduces. When done well, such articles help the community catch up with research debt (Olah & Carter, 2017).

---

[2] https://twitter.com/JayAlammar/status/1331856341397368835
[3] https://twitter.com/maithra_raghu/status/1243551404297342982

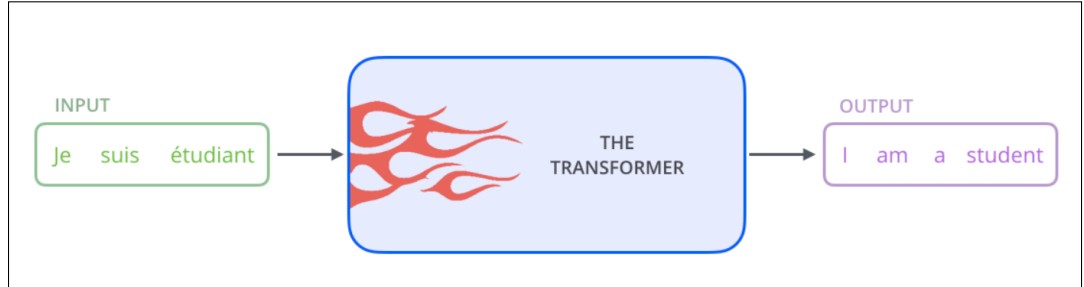

(a) A possible first figure of a model can show the model as a black box and show its inputs and outputs in a way that demonstrate some tangible value.

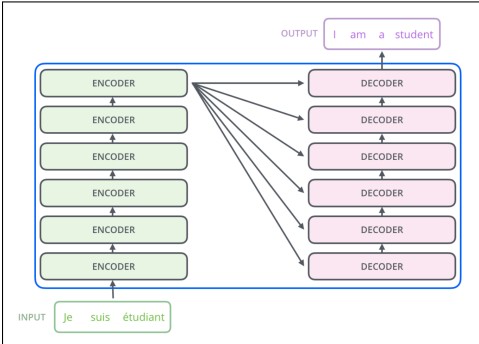

(b) Later figures successively reveal more and more detail, avoiding large mental jumps which may intimidate or distract the reader.

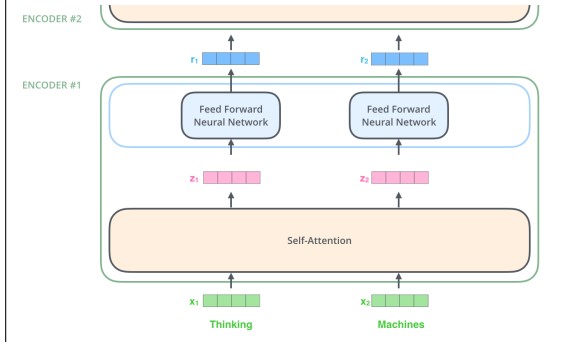

(c) After giving the reader a general sense for the architecture, more information can be revealed about the process, the order of computation, and how the input data flows through the model to produce the output.

Figure 3: Machine learning research can be made more accessible through web articles that communicate concepts with a wider audience in mind. By hiding complexity at first and slowly revealing more and more of a work's key concepts, any writer can help the community by creating gentle on-ramps to ideas they deem important.

**Empathy to the frustration of learning**  Our workflow is most often motivated by the frustration of learning a new concept. It's important to note the details of one's frustration during learning about the concept one aims to explain. Which mental jumps were the most difficult? Which parts could have been explained better? Are there complex parts that could have been better explained using a graphic? Noting one's own emotional state when learning a concept and playing that back at the time of writing helps anticipate the needs and expectations of like-minded readers.

In reality, the cognitive bias known as The Curse of Knowledge (Newton, 1991) often impedes this process, especially for experts who take for granted the underlying concepts of a field they have worked in for years. The next best strategy would be to explain the concept in plain terms to a child and pay attention to when they lose focus or show signs of confusion.

**Intuition first**  Out of the frustration of learning should result in a broad hierarchy of concepts and an appropriate order for explaining them. This hierarchy informs the outline of the article. In our workflow, we find it best to open with a motivating real-world use case whose value can be appreciated by most readers. We present an example in Figure 3. That motivating opening can be followed by a high-level view of the central concept (Figure 3a) of the work and followed by figures successively revealing more detail. The relevant sub-concepts can then be introduced in the same fashion (Figures 3b and 3c).

**Visual language**  A significant portion of our workflow is dedicated to creating visual language. This is a rapid-iteration process that aims to give a visual identity to major concepts and processes involved in a machine learning system. Our experience is that assigning concepts, systems, and

More formally, gradient × input is described as follows:

$$\|\nabla_{X_i} f_c(X_{1:n}) X_i\|_2$$

Where $X_i$ is the embedding vector of the input token at timestep *i*, and $\nabla_{X_i} f_c(X_{1:n})$ is the back-propagated

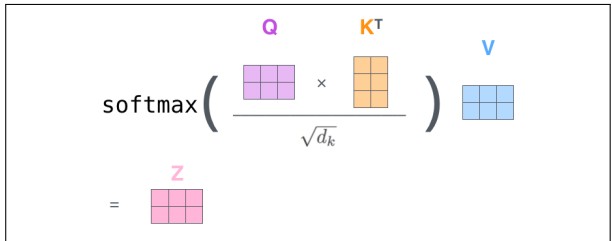

(a) Coloring formulas is a way to link their concept to their discussion in the text and to the explanation of the individual terms

(b) Identifying tensors with colored shapes allows them to be used in formulas – further aiding the reader in grasping how the formula connects to the text and other explanation figures.

Figure 4: Adding color and figures to formulas connects them to the narrative surrounding them.

components their own identifying shapes and colors helps a reader keep track of those elements through the article.

We once again point towards Figure 3 for examples of visual language produced by our workflow. Figure 3a assigns a model a branded color and look. That color is used as the frame in 3b when we look inside the model for more details. The appropriate sub-components are assigned their own colors (Encoders are grouped in one color, decoders in another). Figure 3c explicitly shows tensors and how they factor in the process of calculating a prediction. Keeping track of tensors, their shapes, and relationships is a common challenge across machine learning communication, and one we find is low-hanging fruit when visually explaining machine learning methods.

**Animation**    Web documents allow for the inclusion of animations, which can pack a lot of information about processes in a small area. In our workflow, animations are created in Apple Keynote, then recorded by a screen recorder, then exported to either the Gif, MP4, or WebM format. In machine learning, animations can be invaluable in showing the order of processing inputs, layer activity, and how tensors flow through a model.

**Formulas**    In our workflow, we use the Katex[4] javascript library for mathematical typesetting in web documents. To further aid the reader in comprehending mathematical formulas, we strive towards coloring important terms to visually connect them with their explanation using those colors as can be seen in Figure 4a. We also extend the visual language described above to formulas as can be seen in Figure 4b. Section 2.4 demonstrated adding interactivity to these colored and visualized formulas.

**Pedagogical considerations**    The following are some heuristics we have found helpful in broad communication of machine learning concepts:

- Academic research aims to push the boundaries of human knowledge. The eight-page limitation of the conference paper limits the audience that's possible for the author to address. Explanatory web articles can go a long way in complementing research by making it more accessible to a wider audience.

- Authors of explanatory web articles often intend to create the ideal explanation that can be understood even by the layperson. In practice, this is not easy given the conceptual scaffolding needed to take the general public to the frontiers of knowledge in that space. One possible structure for an explanatory web article is to have three audiences in mind: the general public, specialists (e.g., software engineers, or data scientists), and super specialists (e.g., NLP researchers). The article would then be structured in increased complexity, ensuring that members of each audience learn *something* about the concept. We demonstrate these learning curves in Figure 5.

---

[4]https://katex.org/

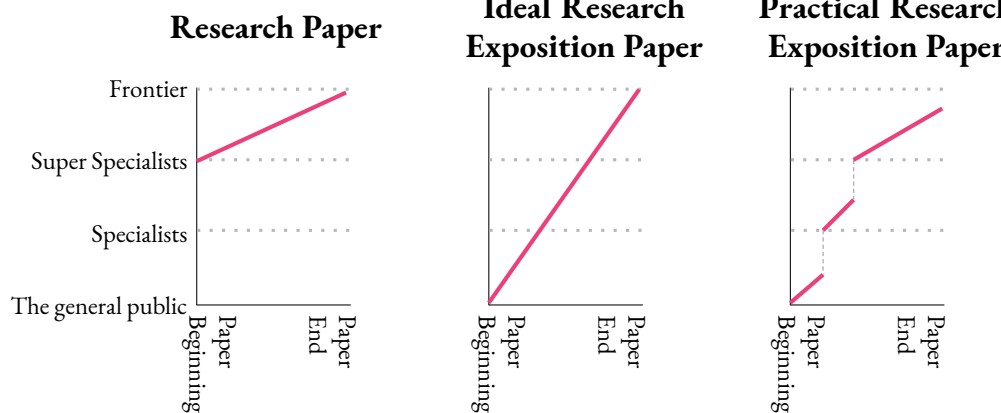

Figure 5: The common research paper communicates a concept on the frontier of human knowledge to a highly specialized audience (left). Explanatory articles aim to explain concepts to the layperson (center); a very difficult proposition for a single article. Practically speaking, having several audiences in mind, and successively addressing them is one possible strategy.

- When explaining a prediction model, it is clearer to start by explaining how a trained model makes a prediction. After that is established, then the training process can be addressed.
- Where possible, machine learning communicators need to avoid cognitively overloading their audience. A common example is when machine learning is introduced using computer vision tasks like classifying images. We advise against this because it puts several non-trivial concepts (i.e., numeric representation of images, pixel values) as blockers between a general audience and the key machine learning concept. Tabular data is perhaps the most user-friendly for general machine learning introductions.

## 2.4 THE INTERACTIVE ARTICLE

The ability to interact with a system gives the reader a better chance of building intuition for how it works (Victor, 2011a). Some interactive articles relevant to machine learning include Victor (2011b) on communicating complex systems, Harris (2014) on K-means clustering, Olah et al. (2018) on interpretability building blocks, Coenen & Pearce (2019) on the UMAP dimensionality reduction algorithm, and Aken et al. (2020) on how the BERT NLP model answers questions. Hohman et al. (2020) provide a broad set of examples and synthesize the medium with relevant theory from domains such as journalism and education.

In our workflow, interactive explorables are built using web technologies like HTML, CSS, Javascript, and SVG. We manipulate these technologies and link them to data using the D3.js[5] Javascript library. One example of our usage of interactivity demonstrates the behavior of a function and how it's calculated as can be seen in Figure 6 of the Sigmoid activation function. More recently, we used interactive explorables to demonstrate NLP explainability techniques (Alammar, 2020).

The web technology stack allows measurement of how users interact with a system as can be seen in Figure 7 which indicates people are fascinated by seeing stochastic gradient descent in action.

Interactive articles are software applications, and so are less durable than a document with images and animations. In our workflow, we rely on client-side scripts that do not require a server back-end to do anything beyond serving static files. While not immune to bitrot, this format is more durable than web applications which require a back-end environment to be maintained and hosted for durations of time potentially exceeding the authors' focus on the problem.

---

[5]https://d3js.org/

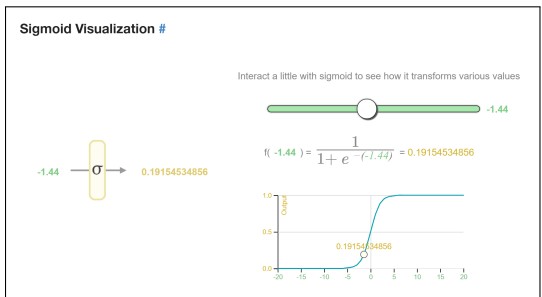

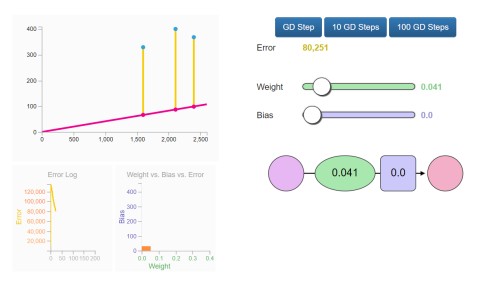

Figure 6: An interactive visualization can convey the behavior of a function in response to changing its input. Formulas can be a part of that animation as well. The example shown here is of the sigmoid function.

Figure 7: Web articles allow for the measurement of how users interact with interactive explorables. For example, readers who interact with this explorable of gradient descent tended to press the GD Step button thirteen times, the 10 GD steps fourteen times, and the 100 GD Steps over *forty* times.

## 2.5 INTERPRETABILITY RESEARCH SOFTWARE

Having created an interactive web article, a natural extension would be to go the extra step in enabling the community to recreate the interactive visualizations themselves on their own system or datasets. This can be as simple as sharing a few lines of code invoking commonly available visualization libraries. While all complex ideas can benefit from visual and interactive explanations, research areas like machine learning interpretability stand to gain a lot from advanced visualization tools.

Notable examples of interpretability research software released as open-source include Olah et al. (2017); Strobelt et al. (2017); Strobelt et al. (2018); Vig (2019); Kokhlikyan et al. (2020); Hoover et al. (2020) and Tenney et al. (2020).

One common way to share research software is through Jupyter notebooks. Even more convenient is sharing notebooks that can run on an online service[6] where the author can ensure the software works in the same software environment the users will run the software in.

**Technical discussions of our software workflow**   As the functionality and size of the software increase, it starts to become a better idea to package that code into its own Python package, which can be distributed online and installed via package managers like Python's pip[7] package installer. Carving functionality out from notebooks and into modules, classes, and functions allows for vastly improving the software by enabling automated testing of individual pieces of functionality.

In creating our NLP explainability package, Ecco[8], our workflow relied on software engineering principles to ensure a standard of quality and aid smoother development and troubleshooting flows. We heavily relied on automated testing (using Tape[9] in Javascript and pytest[10] in Python). Beyond the assertions made in testing the software, our automated tests saved resulting figures and interactive explorables as build artifacts that can be checked to assure new changes to the codebase do not break their functionality. We used Github to host the software repository and run the automated tests after each commit to the repository.

As research becomes software, proper communication begins to include clear software documentation. We documented the software using python docstrings and markdown documents. We config-

---

[6]https://colab.research.google.com/
[7]https://pip.pypa.io/
[8]https://github.com/jalammar/ecco
[9]https://github.com/substack/tape
[10]https://github.com/pytest-dev/pytest

ured Mkdoc[11] to generate static documentation web pages and hosted it on Read The Docs[12]. We organized the documentation using the system[13] breaking software docs in tutorials, how-to guides, technical references, and explanations.

## 3 BIBLIOMETRICS

We were surprised to receive over a hundred citations to the web articles created by this workflow as they were not created inside the academic system. There are few systemic incentives to create this kind of research exposition, so citations are a welcomed token of appreciation from readers who found the work helpful.

## 4 INTEROPERABILITY

Illustrated web articles should maintain the majority of their communicative effectiveness when reduced to a static medium like PDF or print when only animation and potentially color is lost. Interactive web articles face other interoperability challenges where multiple web browsers on multiple operating systems should be tested.

## 5 ACCESSIBILITY STATEMENT

To broaden the accessibility of the visual language discussed in Section 2.3, it's recommended whenever possible to pick colors from perceptually uniform palettes so that colors vary in lightness as well as hue to better accommodate readers with color vision deficiencies.

The underlying motive for improving machine learning communication is to take that knowledge out of the lab and into the minds and hands of as many people as possible. Beyond conceptual accessibility, our commitment to accessibility extends to people of diverse ability levels. We welcome feedback on how we can improve the accessibility of our work on the author's email address.

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
