# OpenReview forum: "Machine learning research communication via illustrated and interactive web articles"
_ICLR.cc/2021/Workshop/Rethinking_ML_Papers/Exhibit_and_Workflow — Rethinking ML Papers - ICLR 2021 workshop Oral_

### Official Review · Reviewer_G96q · 2021-03-26
**Useful Reflection and Insights on Communicating ML Research**

**Accessibility:**

Score of 3 (Neutral): Submission proposes methods to improve accessibility, but the level of intended accessibility is not well-articulated. Also, the limitations and exceptions are not stated.

**Litreview:**

Score of 4 (Strong): The submission directly differentiates itself from previous works and formats.

**Problemstatement:**

Score of 4 (Strong): The submission sets a very strong example of how to address the problem, which should be relevant to the workshop themes.

**Relevance:**

Score of 5 (Exceptional): Like (4) but does so with multiple themes of the workshop.

**Results:**

Score of 4 (Strong): Submission is very well structured and follows all the criteria (i.e. clarity, novelty, interactivity, and coherency). However, practical significance/theoretical implications are not discussed.

**Reviewerconfidence:**

4, I read the article carefully and checked the linked material and examples. I am quite familiar with parallel effort on illustrating machine-learning research, esp. when visual illustration and animations play a central role.

**Reviewtext:**

The author reflects on their extensive experience in communicating ML research to a broad audience, mentioning their sources of inspiration, their operational workflow, and concluding with their observations and lessons learned. I especially enjoyed their take on pedagogical considerations, visual language, and the role of an iconic image (hero image in their words).

While the authors did anchor the importance of their workflow, it would be helpful to have a dedicated discussion on its significance/theoretical implications, e.g., how likely it could be adopted by different groups (educators, practitioners, and researchers), how it could help in correcting any mis-conceptions about ML (both technical and societal), and how it could potentially inform policy makers who need to have a good intuition when regulating the use of ML.

Finally, the accessibility statement felt a bit hand-wavy. It is quite understandable that the author could not address this upfront, however, they were open for feedback form the community. In this regards, narrated animated videos could go a long way to make some of the articles published by the author accessible to readers with motor disabilities, and might be worth adopting as part of the workflow.

Issues with the references:
- Britz: change link to wildml.com/2015/11/understanding-convolutional-neural-networks-for-nlp
- Format all URLs properly (e.g. Britz et al, Nielsen) and unsure they are clickable
- Change visited date to latest possible (e.g. 03/29/2021) for Britz and for Ruder
- Add how published to:
- Devlin: NAACL 2019
- Kokhlikyan et al: arXiv:2009.07896
- Newton: PhD Thesis at Stanford University
- Olah: http://colah.github.io/posts/2015-08-Understanding-LSTMs
- Raghu: arXiv:2003.11755
- Vaswani: NIPS 2017
- Strobelt 2017

Some typos:
- an interactive explorables
- sophestication
- Use consistent capitalization of The in the first paragraph of Section 2.
- Capitalize jupyter, gif, WebM, python (unless you mean the snake)

**Score:**

Strong accept: The reviewer has a strong enthusiasm to apply the proposed framework in their work.

---

### Official Review · Reviewer_DMLC · 2021-03-30
**Improve clarity of scientific communication using illustrations and interactive visualizations**

**Accessibility:**

Score of 4 (Strong): Submission states accessibility concerns and provides solutions within the proposed framework. However, it does not declare the limitations and exceptions.

**Litreview:**

Score of 3 (Neutral): The submission acknowledges previous work, but does not necessarily explain how the submission differentiates itself (i.e we want to avoid the “deluge of citation” strategy, leaving the reviewer to click through references and figure this part out for themselves).

**Problemstatement:**

Score of 4 (Strong): The submission sets a very strong example of how to address the problem, which should be relevant to the workshop themes.

**Relevance:**

Score of 4 (Strong): The submission directly addresses a theme of the workshop, and does so in a very professional manner.

**Results:**

Score of 4 (Strong): Submission is very well structured and follows all the criteria (i.e. clarity, novelty, interactivity, and coherency). However, practical significance/theoretical implications are not discussed.

**Reviewerconfidence:**

Score: 4

The author has presented good techniques on how one can improve the readability of their research paper and how to communicate better. More illustrations and visual elements definitely helps in understanding complex concepts mentioned in a research work.

**Reviewtext:**

This paper presents a workflow on how to improve the clarity on how research ideas should be communicated in research papers, promote interpretability, and empower the community to reproduce research work. Author's workflow is described using different communication artifacts like using:
1.  Hero image: An image that explains a central concept of a research work aiding understanding and recall of the paper
2. Twitter thread: A twitter thread to motivate the concept and use an image to demonstrate the accessible visual style used in the remainder of the thread
3. Animated web article: Series of images and animations explaining a machine learning concept to a wide audience
4. Interactive article: A demo of the research work with the focus on making sure that it is a client side application and no backend involved so that the interactive article has a longer life
5. Interpretable research software: Interactive visualizations on the original datasets on jupyter notebook or google colab like setup.

Strengths:
- The paper is very well written and explains well about how we can improve the clarity of research work so that the audience understands it really well.
- The author also focuses on how one should make their teaser figure (the hero image) which is one of the most important pieces of the paper these days in machine learning research.
- Section 2.3 lays down down a really good plan on how one should decide on the figures and what colors to pick for the different figures (making sure that one is following a consistent color scheme across different figures for better understanding)

Weaknesses:
- Minor: Although the author has really good proposals, it would have been great if they could have done some human studies on how these interactive tools and visualizations help in improving the research communication.

**Score:**

Accept: The reviewer believes the submission provides a novel and reliable scheme to improve science communication but needs improvement.

---

### Meta-Review · Area_Chair_wSRp · 2021-03-31

**Recommendation:** Accept
**Confidence:** 4

**Metareview:**

This exhibit/workflow submission presents a set of design principles behind a very sucessful set of web/interactive articles. These principles are worth discussing in the workshop, and will perhaps find early adoption (or equally, constructive criticism).

The rationale behind the concepts of a hero image, a Twitter thread, or a communicative article is clearly outlined. This article also touches upon pedagogy and interpretability-related nuances (such as colored visualization of equations and associated textual definitions).

However, as pointed in one review, this article  could discuss other similar efforts towards interactive articles (Distill.pub, CodaLab, etc.). Further, the scope of the accessibility statement could be broadened to include guidelines for web / interactive articles (currently, it focuses on the visual language contributions). As a concrete example, alt-text should be encouraged (perhaps enforced) for images in web articles.

In summary, this exhibit/workflow makes interesting and well-founded contributions along the primary themes of the workshop. I recommend inclusion of this exhibit/workflow into the workshop proceedings.

---

### Decision · Program_Chairs · 2021-04-01

Accept (Oral)